# Earthquakes on the surface: earthquake location and area based on more than 14500 ShakeMaps

Stephanie Lackner[1]

[1]Woodrow Wilson School of Public & International Affairs, Princeton University, Princeton, NJ, 08544, USA

*Correspondence to:* Stephanie Lackner (slackner@princeton.edu)

**Abstract.** Earthquake impacts are an inherently interdisciplinary topic that receives attention from many disciplines. The natural hazard of strong ground motion is the reason why earthquakes are of interest to others than seismologists. However, earthquake shaking data often receives too little attention by the general public and impact research in the social sciences. The vocabulary used to discuss earthquakes has mostly evolved within and for the discipline of seismology. Discussions on earthquakes outside of seismology thus often use suboptimal concepts that are not of primary concern. This study provides new theoretic concepts as well as novel quantitative data analysis based on shaking data. A dataset of relevant global earthquake ground shaking from 1960 to 2016 based on USGS ShakeMap data has been constructed and applied to the determination of past ground shaking worldwide. Two new definitions of earthquake location (the shaking center and the shaking centroid) based on ground motion parameters are introduced and compared to the epicenter. These definitions are intended to facilitate a translation of the concept of earthquake location from a seismology context to a geography context. Furthermore, the first global quantitative analysis on the size of the area that is on average exposed to strong ground motion - measured by peak ground acceleration (PGA) - is provided.

## 1 Introduction

Earthquakes receive a lot of attention from the general public as well as numerous disciplines across the natural and social sciences. With the notable exception of seismology, most of them are primarily or even exclusively concerned with the surface phenomenon and the impacts of earthquakes. However, the literature commonly uses magnitude or other suboptimal measures to quantify the natural hazard of earthquakes for impact research. The physical phenomenon of strong ground motion does often not receive enough attention and the literature lacks an interdisciplinary discussion of the natural hazard of earthquake-related surface shaking.

Earthquake risk communication is generally considered a high priority topic and many research and practical efforts are concerned with educating the public and improving preparation. The Southern California Earthquake Center (SCEC), for example, has started the Great California ShakeOut (Jones and Benthien, 2011) which has become an annual drill with millions

of participants, the Global Earthquake Model (GEM) project is an international effort to develop a global model of earthquake risk as an open source, community-driven project (Crowley et al., 2013, www.globalquakemodel.org), and the Global Seismic Hazard Map project (GSHAP) has promoted a regionally coordinated, homogeneous approach to future seismic hazard evaluation, including the production and distribution of a global seismic hazard map (Giardini, 1999). Technological progress has also allowed for the emergence of real-time seismology, which provides real-time information about an event during and in the immediate aftermath of an earthquake (Kanamori, 2008).

For research on earthquake impacts, an appropriate understanding of the physical hazard of past earthquake shaking as well as access to relevant data is necessary. However, earthquake communication about past events is a relatively neglected topic and many authors have to struggle with the inconsistent - and sometimes inadequate - approaches in the social science literature (Kirchberger, 2017). This study utilizes USGS ShakeMap data (Wald et al., 1999) - a real-time seismology product - to create a dataset of global past shaking exposure. Natural hazard exposure maps are necessary for impact research, and they also allow to spatially overlap the natural hazard with social variables representing vulnerability or preparedness. This study provides a discussion and quantitative analysis of global earthquake ground shaking and variables of interest that can be calculated from such data.

## 2 Shaking Data

Many different factors about an earthquake play important roles in what kind of shaking is experienced on the Earth's surface. Douglas (2003) classifies them into three categories: those related to the earthquake source (e.g. magnitude, depth, or faulting mechanism), travel path (e.g. geology can have a significant impact on attenuation), and local site conditions. The prediction, estimation, and recording of strong ground motion parameters are active fields of ongoing research and technological improvements (Douglas, 2003; Denolle et al., 2014; Wald et al., 2011; Caprio et al., 2015; Kong et al., 2016). SCEC, for example, has a Ground Motion Prediction Working Group.

Strong ground motion can be expressed with different parameters, but it is commonly characterized by peak ground acceleration (PGA), peak ground velocity (PGV), and peak ground displacement (PGD). More sophisticated measures such as response spectra or Arias intensity (Arias, 1970) generally provide a better characterization of strong ground motion (Joyner and Boore, 1988), but such data are not as easily available as for PGA or PGV. No individual index of strong ground motion is ideal to represent the entire frequency range, but peak ground motion parameters are considered to perform satisfactorily (Riddell, 2007).

This study will focus on PGA as the ground motion parameter. However, PGA alone only provides a limited representation of ground motion. To represent the entire frequency range more appropriately a multi-parameter characterization of ground motion is commonly used (e.g. for selected earthquakes the ShakeMap product also provides response spectra maps at periods of 0.3 s, 1 s, and 3 s - according to three Uniform Building Code reference periods). Nevertheless, while appropriate in engineering, a multi-parameter approach is not reasonable for many social science applications. PGA is still widely use in

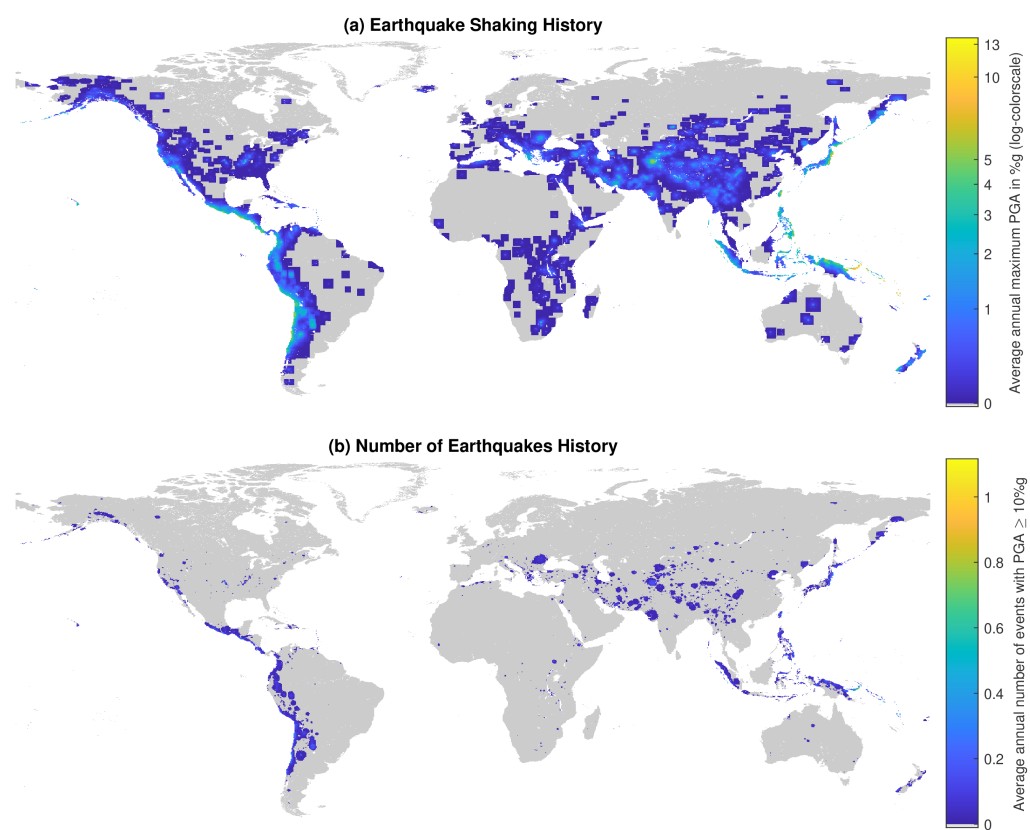

**Figure 1.** Earthquake shaking history of 1973-2015. Panel (a) shows the average annual maximum shaking exposure and panel (b) provides the average annual number of "big" shaking events by location. Panel (a) also illustrates some limitations of the data, such as the edges of individual ShakeMaps and an over-representation of low shaking events in more populated regions (e.g. Australia). It is also important to be aware that the shaking history is a combination of observed and estimated data for actual shaking exposure in the given time period.

earthquake engineering, and it provides the advantage of a single-valued parameter with good data availability. Expressing past shaking with PGA is also consistent with the common approach of using PGA for earthquake hazard maps.

While more sophisticated approaches to calculate ground motion parameters (e.g. those employed by SCEC) than the ShakeMap methodology exist, USGS ShakeMaps are unique in providing consistent earthquake strong ground motion data for a large number of events on a global scale and for several decades. For this study 14608 ShakeMaps of earthquakes from 1960-2016 have been compiled into one dataset. Each ShakeMap consists of observed instrumental ground motion parameters where available and estimates from models based on ground motion prediction equations (GMPEs), nearby observations, and other data, where no observations exist. The ShakeMap methodology is continuously improved and documented in numerous publications which can be found through the USGS website (https://earthquake.usgs.gov/data/shakemap/background.php).

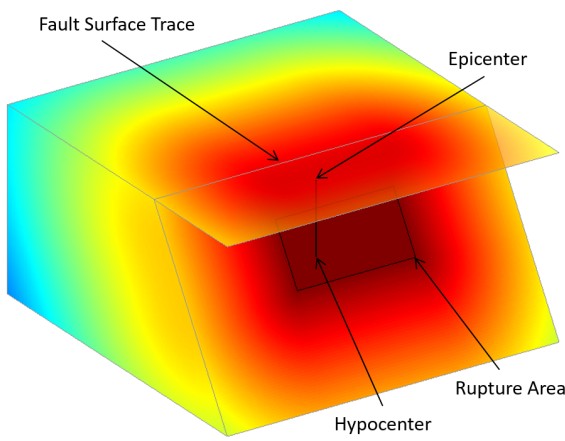

**Figure 2.** Illustration of a fault plane and wave intensity. The earthquake nucleates at the hypocenter, but waves radiate out from every point of the rupture area. For this reason as well as other factors (e.g. local site conditions and water bodies), the epicenter is not necessarily in the center of the strong shaking area.

In December 2016 all online available USGS ShakeMaps were collected and combined into one dataset for this study. The ShakeMaps were combined with the GPWv4 gridded Land and Water Area (Center for International Earth Science Information Network - CIESIN - Columbia University, 2016) to restrict the ShakeMaps to only on surface land shaking. Except for one event in 1923, ShakeMaps generally exist starting from 1960, and they are systematically available from 1973 onwards. The representativeness of the sample of earthquake ground shaking in the dataset compared to all earthquake ground shaking in the time period is assessed by matching the ShakeMap data to two different earthquake lists, which allows a cross validation. Since not every earthquake has a ShakeMap, a reference dataset of all (or at least almost all) earthquakes since 1960 is required. For this purpose the ANSS Comprehensive Earthquake Catalog (ComCat, 2017) is used. More information on the ComCat data can be found in Appendix A. Finally, the NGDC Significant Earthquake Database (National Geophysical Data Center / World Data Service (NGDC/WDS), 2017) is applied as an additional reference dataset and to assign impacts to events (the impact data will be used in future research). The combination of the three earthquake data sources can help to identify how representative the aggregated ShakeMap dataset is for all global earthquake ground shaking. Details on how the three datasets were linked with each other are described in Appendix B.

As a result, the dataset can be considered to contain all relevant global earthquake ground shaking from January 1973 to October 2016, and also a reasonable sample from 1970-1972 as well as shaking from individual devastating events from 1960-1969. However, the sample is more complete in later years, in that it contains more weaker events. For smaller events the dataset has a bias towards North American events, which can be avoided by restricting the sample to only events with a magnitude of 5.5 or greater. However, reducing this threshold to 4.5 is generally sufficient to avoid this bias. Details on the representativeness of the ShakeMap dataset can be found in Appendix C.

## 3 Past Earthquake Shaking

Past earthquake shaking can be approached from two different angles, either comparing different locations or comparing different events. First, we can consider the exposure of a location or region to past earthquake shaking of an individual event or to several events over a time period. The constructed dataset allows to visualize the global past shaking exposure. Hsiang and Jina (2014) term the average annual pixel exposure to maximum cyclone wind speeds as the *cyclone climate*. This representation of past exposure to a natural hazard can be particularly useful in social science applications. In a similar way, the average annual pixel exposure to maximum PGA could be called the *earthquake shaking "climate"*. Since earthquakes are not a climate phenomenon and past exposure maps are not equivalent to future hazard maps, it is more reasonable to refer to it as the *earthquake shaking history*. Figure 1 illustrates the earthquake shaking history of 1973-2015, the period for which the dataset is found to be representative of overall shaking in the specific time range. However, some limitations of ShakeMaps become apparent in Figure 1 panel (a): (i) the cutoff edges of the individual ShakeMaps are visible, (ii) some unrealistically high outliers might skew individual pixels, and (iii) low shaking/impact events are more likely to receive a ShakeMap when they are in areas of interest (e.g. cities in Australia). The visualization in Figure 1 panel (b) avoids these limitations by restricting to shaking above a PGA of 10 %g and showing the number of events above that threshold. Earthquake hazard maps are a common way to illustrate where and what strength of future earthquake shaking is likely to occur. Such maps are usually expressed in PGA that is expected to be exceeded at a certain likelihood within a given number of years (e.g. Shedlock et al. (2000)). Using the maximum shaking over time instead of the average annual shaking allows to compare the actually experienced shaking (or estimates of it) with the probabilistic estimates from hazard maps. To illustrate this potential application of ShakeMap data, a comparison of the earthquake maximum shaking history with the GSHAP global earthquake hazard has been conducted and can be found in Appendix E.

Second, the comparison of different past events can be the main objective instead of comparing individual regional (or single coordinate) exposures. Two crucial aspects of a particular earthquake are the location and the area affected by the shaking. It is, however, not straightforward to define these concepts. An earthquake is caused by the rupturing of a fault segment (illustrated in Figure 2). The earthquake originates at the hypocenter, but waves radiate out from the whole segment of the fault that ruptures (rupture area). This results in the epicenter not being necessarily at the center of the strong ground motion area. In the figure this depends on the size of the rupture and the dip angle of the fault. Other factors such as local site conditions and water bodies also affect where strong ground shaking occurs. From a social science perspective the surface projection of the rupture area could actually be considered to be more relevant than the epicenter - which is the surface projection of the hypocenter. However, there are numerous other factors that influence surface shaking and even two earthquakes on the same fault can behave very differently due to their underlying rupture processes.

## 4 Earthquake location

Earthquake location is currently primarily discussed within and from the context of seismology. However, a translation to a geographic language would often be beneficial. Emergency response, hazard management, and regional planning often rely

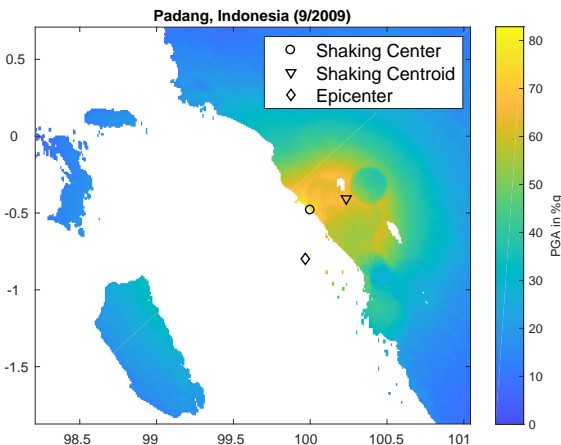

**Figure 3.** An example of a map of peak ground acceleration based on USGS ShakeMap data. The figure shows the locations of the epicenter, the shaking center and the shaking centroid. The shaking center in this example is 35 km away from the epicenter. The distance between the shaking centroid and the epicenter is even 53 km.

heavily on geographic parameters and the use of geographic information systems (GIS). Tobler (1970) coined the "first law of geography" stating that "everything is related to everything else, but near things are more related than distant things". It is therefore not surprising that the location is an information about an earthquake that is of great interest to the general public and disaster management. Furthermore, the consideration of spatial effects (e.g. spatial autocorrelation and spatial heterogeneity)
can be crucial in econometric models (Anselin, 2007) in the social sciences and requires to assign a location to each observation. Thus earthquake location can also be a crucial parameter in social science applications.

The currently most commonly used and calculated points to characterize earthquake location are the hypocenter and the epicenter. However, from a strong ground motion perspective they are not the most interesting points. The epicenter is not necessarily a good proxy for where strong ground motion occurs, and it is thus not the optimal location choice for many
applications. When the epicenter is offshore it can also be far away from the strong shaking region, and it is not straightforward to assign the event to a country or region. Another earthquake location is the centroid location, such as those for example calculated by the CMT project (Ekström et al., 2012). The centroid location is the average location in space and time of the seismic energy release. However, the centroid location does also not account for water bodies and its surface projection can thus often be far from the strong shaking area as well.
In disaster management and planning as well as social science applications, the desired location parameter should summarize the spatial component of earthquake ground motion. While location information about earthquakes is often supplemented with qualitative statements (e.g. "the epicenter is X km offshore" or "the most affected region is X km south of city Y"), the additional information does not necessarily enhance the digestion of information. This is particularly the case when the information provides details about the complexities of a rupture process. Details such as rupture length or directivity are

important aspects about an earthquake in seismology and earthquake engineering, but they do not necessarily facilitate a better understanding for individuals without the respective backgrounds and can even contribute to confusion.

A simple geographic parameter that summarizes the shaking of an earthquake can facilitate a translation from seismology and engineering to a geography context. Since many decision makers are familiar with such a geography context, this can enhance
digestion of information by relevant individuals and groups. Moreover, in some social science applications it is essential to be able to assign coordinates to an individual event. Since the phenomenon of interest in these applications is ground shaking, the purpose of this location is to summarize shaking. So far, no formal definitions for earthquake locations based on ground motion parameters exist. This study will introduce two surface points other than the epicenter, which can both be considered different definitions of the earthquake surface location: the shaking centroid and the shaking center. Both definitions are formulated such
that they can be applied to any ground motion parameter. Nevertheless, this study specifically applies them to ShakeMap PGA data.

The *shaking centroid* $(x_{SCt}, y_{SCt})$ will be defined as the average location $(x_i, y_i)$ of shaking $s_i$ weighted by the squared shaking for a given ground motion parameter, only including locations that experience at least 50 % of the maximum shaking $s_{max}$ of that event.

$$(x_{SCt}, y_{SCt}) = \frac{\sum_{i \in \{i : s_i \geq 0.5 s_{max}\}} (x_i, y_i) s_i^2}{\sum_{i \in \{i : s_i \geq 0.5 s_{max}\}} s_i^2} \tag{1}$$

Restricting the included locations to the area with at least 50 % of the maximum shaking is chosen for two reasons. First, it helps to avoid the problem that ShakeMaps are usually cutoff before the shaking has completely attenuated. Second, it ensures that the shaking centroid represents a location that summarizes best the strong shaking area of the particular event. The weaker shaking area is generally of less interest. The squares of the ground motion parameter are also chosen to allow for a stronger
weight of the high shaking locations. Just like the epicenter, the shaking centroid could be a location that doesn't actually experience any shaking (when it falls in water) or only relatively low shaking itself.

On the other hand, the *shaking center* is the point on the surface which experiences the strongest shaking for a given ground motion parameter.

$$(x_{SC}, y_{SC}) = \{(x_i, y_i) : s_i = s_{max} := max_i s_i\} \tag{2}$$

The calculation of the shaking center provides a challenge when a ShakeMap has more than one location that share this maximum value. Details on the here applied approach to handle this issue are described in Appendix D.

Figure 3 provides an example of a ShakeMap with the three different surface locations. The shaking center and shaking centroid generally do not coincide with the epicenter. In particular when the epicenter lies in water, it will definitely be distinct from the shaking center and it will very likely also be distinct from the shaking centroid. The problems with the definitions
of the shaking center and the shaking centroid are that (i) they depend on the choice of a ground motion parameter, and (ii) any map of a ground motion parameter - and therefore also the shaking center and the shaking centroid - can not be as accurately evaluated as the epicenter. However, the shaking center and shaking centroid are locations of greater interest for many applications.

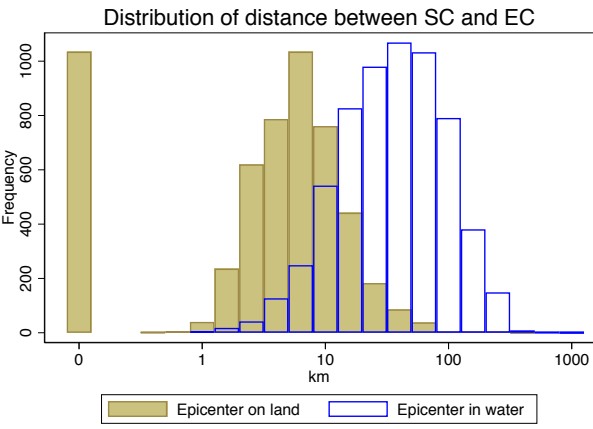

**Figure 4.** Comparing the epicenter (EC) and the shaking center (SC) in terms of distance for all 11510 earthquakes with shaking and magnitude 4.5 or greater. In only about 9 % of earthquakes does the epicenter coincide with the shaking center.

Table 1 compares the locations of epicenter, shaking center and shaking centroid for PGA as the ground motion parameter for all 12388 ShakeMaps in the dataset with magnitude 4.5 or greater. About 57 % of those earthquakes have their epicenter in water and some of those events do not cause any shaking. Among the 11510 events that do cause shaking about 54 % have their epicenter in water. For the 46 % of those earthquakes, which have their epicenter on land, the average distance between the

epicenter and the shaking center is 7 km. This distance increases to 53 km when the epicenter is in water. The full distribution of shaking center to epicenter distances is shown in Figure 4. The PGA at the epicenter is on average 13 % weaker than at the shaking center, given that the epicenter is on land and experiences any shaking (5285 events).

## 5   Strong ground motion area

In terms of the area affected by an earthquake, the literature so far has mainly referred to the area exposed to certain levels of

a qualitative intensity scale for individual events. There has been no study on the global pattern of the area that is on average exposed to strong ground motion parameters for a given earthquake. This study provides the first summary of global earthquake area size.

The size of the area that experiences strong ground motion from an earthquake is strongly dependent on the regional geology. Figure 5 illustrates how different the size of the area exposed to strong ground motion can be across the world. In particular it

shows the average size of the area that was exposed to at least 90 % of the maximum PGA within each grid cell for the time period 1973-2015 (see Figure C1 in appendix C for reference of how many earthquakes are used in each grid cell to calculate the average). For example, earthquakes along the west coast of South America can generally be felt at far wider distances from the epicenter than earthquakes on the west coast of North America. The west coast of South America in Figure 5 also illustrates how earthquakes close to the coast are spatially smaller, since the ocean restricts the shaking pattern to one side. Water bodies

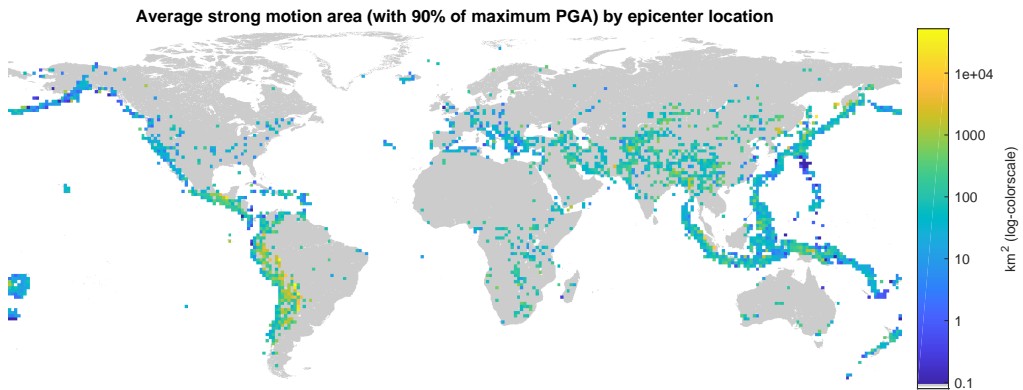

**Figure 5.** Attenuation across the world illustrated by average strong ground motion area for 1973-2015. This map shows the average area that was exposed to at least 90 % of the maximum PGA for an earthquake with the epicenter at that location. For each 1.25 x 1.25 degree grid cell the average area is calculated for all earthquakes of the 12388 ShakeMaps with magnitude 4.5 or greater, which have their epicenter in that grid cell. The total number of earthquakes per grid cell varies significantly and the fact that many regions have recurrence intervals of more than 43 years for strong events makes this map particularly sensitive to the time interval of the data.

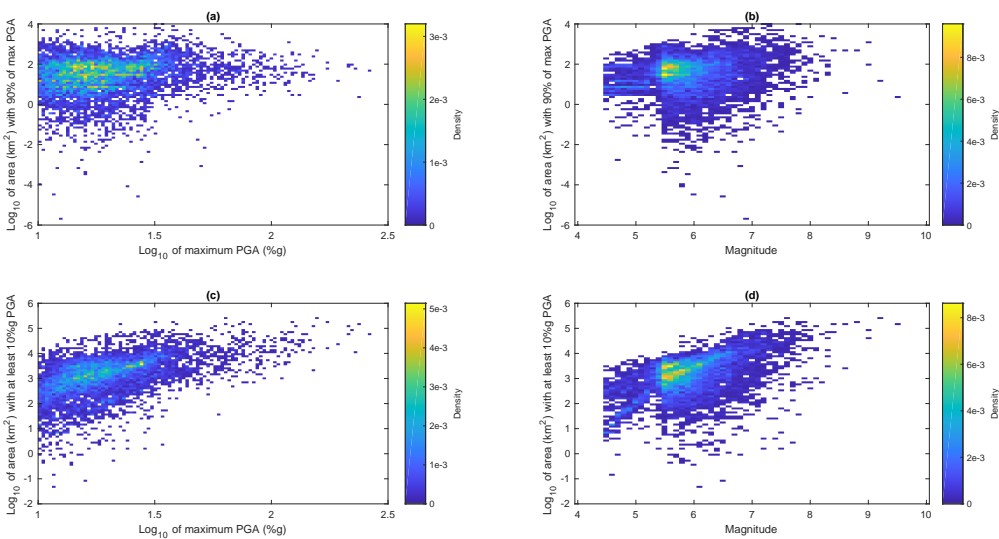

**Figure 6.** Earthquake area compared to magnitude and maximum PGA. The four panels in this figure show scatter plots (with scatter density illustrated by color) to illustrate the relationships between these measures. Only earthquakes with magnitude of at least 4.5 and maximum PGA of at least 10 %g are included.

are crucial in defining the area that can experience ground motion, and therefore also for the area exposed to strong ground motion.

Earthquake magnitude and distance are two of the most important factors in ground motion prediction equations. It is therefore intuitive that magnitude affects the area exposed to a particular shaking threshold, and it is indeed easy to see this relationship in the data. However, as panel (d) in Figure 6 shows, other factors (e.g. geology and water bodies) introduce significant noise in this relationship and make it thus less straightforward. This highlights the importance of other factors than magnitude in determining surface shaking. A more detailed summary of average shaking areas can be found in Tables 2 and 3. The tables provide the average area exposed to 90 % of the maximum PGA for each ShakeMap and the average area exposed to at least 10 %g PGA, separately by magnitude and maximum PGA level. The total number of earthquakes in each category that is used to calculate the average can be found in Table A1.

A stronger magnitude event - keeping everything else about the earthquake constant (i.e. depth, geology, fault type, hypocenter location, etc.) - will result in larger areas exposed to any given PGA level. This is generally confirmed by the data in Table 3. A high magnitude event can only have a low maximum PGA level when the epicenter is in water. It is, however, likely that a still large area would be exposed to these relatively low shaking values. Such events are responsible for the very large areas exposed to 90 % of the maximum PGA in Table 2 and panel (b) of Figure 6. While an increase of the area above a fixed PGA threshold with increasing magnitude is intuitive, Table 2 and panel (b) of Figure 6 suggest that also the area exposed to a fixed percentage of the maximum PGA increases with magnitude. This is most likely due to the fact that large magnitude events tend to turn the large amount of energy released not necessarily into stronger shaking, but into larger areas experiencing strong shaking. A similar relationship does not seem to hold for maximum PGA and area (see panel (a) in Figure 6). This can be interpreted as the size of the area exposed to a certain percentage of the maximum PGA being independent of the maximum PGA, but dependent on magnitude. Earthquake magnitude therefore seems to contain more information about the spatial extent of an earthquake than the maximum PGA. The relationship between magnitude and the area exposed to at least 90 % of the maximum PGA could potentially affect the global pattern of attenuation illustrated in Figure 5. However, the same figure only for earthquakes with magnitude between 5.5 and 6.5 (see Figure C2) confirms the overall pattern.

## 6  Conclusion

This study provides a discussion of earthquake shaking data for an interdisciplinary audience and with applications in earthquake impact research, particularly in the social sciences in mind. It constructs and utilizes a comprehensive dataset of global strong ground motion data to define new concepts of earthquake location as well as strong shaking area that help summarize the natural hazard of surface shaking. These concepts can help to facilitate a more effective communication about the natural hazard of past earthquakes that is focused on surface shaking. The concept of a shaking center and a shaking centroid are introduced, which can often be better suited location definitions for an earthquake than the epicenter in social science applications as well as disaster management.

More than 14500 individual ShakeMaps were compiled into one comprehensive dataset. The dataset can be considered to contain all relevant global earthquake ground shaking from 1/1973 to 10/2016, and also a reasonable sample from 1970-1972 as well as shaking from individual devastating events from 1960-1969. Using observed or estimated shaking data of past events can be used to compare hazard maps with maps of actual shaking occurrences. An example of this application can be found in Appendix E, which compares the maximum PGA exposure for 1973 - 2015 according to the ShakeMap data with the GSHAP hazard map of probabilistic estimates.

The dataset is applied to calculate the shaking center and shaking centroid for all events in the dataset. The shaking center is particular useful to assign a country to an event, since it is always on land. The shaking centroid, on the other hand is generally the best representation of the overall location of shaking. It is the most reasonable choice for the assignment of an event to a general region or to use as the location in spatial regression models or other statistical tools. The CMT centroid location could also be a relatively good predictor of the location of strong surface shaking. An interesting future extension of this research would therefore be to combine the ShakeMap dataset with the CMT data to compare the shaking center and shaking centroid with the CMT centroid location.

Finally, the dataset is also applied to calculate a number of different shaking area variables. This work provides the first summary of global earthquake strong ground motion area size. The average strong ground motion area is shown to be a useful tool to visualize attenuation across different regions of the world.

The constructed dataset can be used in future research on determining the short-term and long-term impacts of past earthquakes. This will allow to investigate the impacts of earthquakes based on measures that represent the natural hazard of interest: earthquake ground shaking.

*Code and data availability.* The data sources used in this study are freely available at the USGS website (https://earthquake.usgs.gov/data/shakemap/ and https://earthquake.usgs.gov/data/comcat/), the NGDC website (https://www.ngdc.noaa.gov/hazard/earthqk.shtml), and the GSHAP website (http://static.seismo.ethz.ch/GSHAP/global/). The code that was used to process the data and create the tables and figures, as well as more details about the raw data are available at the following GitHub repository: https://github.com/slackner0/EQSurface.

### Appendix A: ANSS Comprehensive Earthquake Catalog (ComCat)

For this work the list of all events with magnitude 4.5 or higher that occurred between January 1960 and October 2016 is used. Additionally, events below magnitude 4.5 during that time period were added if they do have a ShakeMap according to ComCat. This results in a list of 225429 earthquakes. The data was downloaded in March 2017 with the ComCat online access tool (https://earthquake.usgs.gov/earthquakes/search/). The threshold of 4.5 is chosen since earthquakes outside the US below this magnitude are not as systematically recorded in ComCat. We thus have a reliable but not entirely complete list of global earthquakes of magnitude 4.5 and higher for the chosen years. Since some of the earthquake data sources in ComCat

only provide data starting from certain years, the data is more complete for more recent years. As the analysis will show, particularly for the time period 1960-1972, the ComCat list can not be considered complete.

A more likely problem than the lack of events in the list, are possible duplicate events in the list. Earthquakes often occur in clusters. A big event might have foreshocks or aftershocks. Sometimes two different earthquakes occur at very close proximity and less than a minute apart. However, a close investigation of the ComCat list reveals that some of those particularly similar events in terms of timing, location and magnitude, might actually not be separate events, but the same event with slight differences in the estimated source parameters from different data contributors. I exclude 33 events from the gcmt network for which another event within 1.5 seconds at a distance of under 3 degrees exists, since those events seem to be duplicates. Furthermore, I also exclude 6 events that don't fulfill these criteria, but have been manually identified as most likely duplicates. There are, however, most likely more duplicate events as the representativeness analysis shows. After excluding these events, we have a list of 225390 events which can be uniquely identified by the combination of the following parameters (rounded to specific accuracies in parentheses): timing (to the minute), magnitude (.1), longitude (1), latitude (.25), and depth (25).

## Appendix B: Linking the three earthquake datasets

The first step is to combine the ComCat earthquake list with the ShakeMap dataset. Unfortunately, the ComCat list and the ShakeMap data are often updated separately from each other and earthquake source parameters (e.g. magnitude, timing, location) can therefore differ between a ComCat event and the ShakeMap for the same event. Also the earthquake "ID" does not always agree between the ShakeMap and the corresponding ComCat event. The differences in source parameters either stem from the data providing network updating the parameters or from different networks being chosen for the ShakeMap and the ComCat with slightly deviating parameters. Sometimes also the magnitude type might be different, resulting in different magnitude values. When possible events from the datasets are matched by timing (to the minute), magnitude (rounded to .1), longitude (rounded to 1), latitude (rounded to .25), and depth (rounded to 25). Such a match is possible for 7882 ShakeMaps.

The remaining ShakeMaps are matched to the remaining ComCat events (i) if they are at most 60 seconds apart, at a (euclidean) distance of at most 2 degrees, and have a difference in magnitude of at most 2.2 (0.7 if the ShakeMap magnitude is below 5.5), or (ii) if they occur within 2 seconds and at a distance of at most 2 degrees. If several events fulfill these criteria, the event with the least time difference and the event with the least spatial difference are identified, and if they are the same event it is assigned to the ShakeMap. Otherwise the event with the least time difference is chosen if that time difference is at most one fifth of the next closest event (in terms of timing). If that again is not the case, the spatially closest event is chosen, given that it has a spatial distance of at most 1 degree. For all so far unmatched events of relevance (high magnitude of the ShakeMap or ComCat event, or ComCat indicates that a ShakeMap should exist for an event) a manual check and potential assignment is done. For 20 events a manual assignment was necessary to match the right ComCat event and ShakeMap.

This process finally results in a total of 14592 ComCat events with ShakeMap. According to ComCat only 5310 events are supposed to have a ShakeMap. It was, however, possible to find significantly more than that on the USGS website. Nevertheless there are 127 events which are supposed to have a ShakeMap, which is missing in the dataset. Most of them are missing because

they were produced after December 2016 when the ShakeMaps were downloaded and some also because the ShakeMap files were corrupted. The magnitude of 67 of those events is below 4.5 and for only three of the 127 events is the magnitude higher than 5.5. It is therefore reasonable to assume that the exclusion of these 127 ShakeMaps will not affect the representativeness of the dataset in a significant way.

The second step is assigning each event from the significant earthquakes list to a ComCat event. Again, the source parameters show slight deviations and a similar approach as matching ShakeMaps with ComCat events is utilized. Each significant earthquake event is matched to a ComCat event if they are at most 90 seconds apart, at a distance of at most 5 degrees, and have a difference in magnitude under or equal to 2. If more than one ComCat event fulfills these criteria the event with the smallest time difference and spatial distance is chosen (they always agree for this dataset). However, for some events the significant
earthquake list has missing timing data (second, minute, or hour). For those events, the timing has to be within the same day and the spatial difference can not be more than 2.5 degrees. Additionally, 14 events were matched manually. Unfortunately some events in the significant earthquake list seem to have typos (e.g. a drop in the leading 100 of a longitude location). Some typos are identified manually and they are part of the 14 manual matches, but there are potentially more typos or just deviations in the data in terms of the timing. For all unmatched events with no ComCat event within 90 seconds, we therefore identify
matches if they are within 24 hours, at a distance of at most 0.2 degrees and have a magnitude that deviates by at most 0.2.

    All but 152 of the 2130 significant earthquakes can finally be matched with ComCat events. An additional 16 events of the significant earthquake list that are not in the ComCat list were able to be matched with yet unmatched ShakeMaps. It is unlikely that excluding the remaining 136 events will bias the data in a problematic way. First, most of those events have relatively low magnitudes and are therefore not in the ComCat list (114 of the 136 events have a magnitude below 5.5). Second, 86 of the
136 earthquakes stem from the period 1960 - 1972. This is a sign that the ComCat list for that period is not as complete as for later periods, which we already expect from the data availability of the ComCat data sources. Finally, considering the impact of fatalities, 91 of the 136 events caused at least one death, but the average among those is only 11, with a maximum of 80. After 1972, the largest number of fatalities among these events is 14.

    For the time period of January 1960 - October 2016 we have 14608 ShakeMaps that are either matched to a ComCat event
(13061), or a significant earthquake list event (16), or both (1531). For those events we will use the source parameters from the ShakeMap and disregard the potentially deviating ComCat and significant earthquake list parameters. If no ShakeMap exists, the ComCat source parameters will supersede the significant earthquake list parameters in the dataset.

**Appendix C: The representativeness of the ShakeMap dataset**

The combination of the three earthquake data sources can help to identify how representative the aggregated ShakeMap dataset
is for all global earthquake ground shaking. The first concern is whether "big" events - either in terms of shaking or impacts - might not be in the dataset. The ShakeMap creation criteria (Allen et al., 2009) are supposed to ensure that this doesn't happen. However the significant earthquake list provides us with a tool to test this. We would generally expect that a "significant" earthquake should have a ShakeMap. Indeed, 1547 significant earthquakes in our combined dataset do have a ShakeMap. We

already discussed the 136 cases of significant earthquake events that could not be matched to either a ComCat event or a ShakeMap. Those events are either relatively small, or they are from the time period 1960-1972, suggesting that the ComCat event list is not complete for that time period. Of the 447 significant earthquake list events which have been matched to a ComCat event without a ShakeMap, 99 caused at least one fatality. This is a concern, since events with fatalities should usually

have a ShakeMap. However, 84 of those events are from the time period 1960-1972. This is not unexpected, since ShakeMaps are not systematically produced before 1973. The remaining 15 events after 1972 with fatalities but no ShakeMap have on average 3 fatalities and a maximum of 11. We can therefore expect that these events are sufficiently small to not miss a major impact event. Of the entire 447 events, 207 are from 1960-1972 and they cause therefore no additional concern beyond the already known unreliability of that time period. For the remaining 240 events, only 73 have a magnitude of 5.5 or higher. Many

of these higher magnitude events are in remote locations such as Antarctica or Alaska, or occurred far offshore and did not cause a lot of shaking. We have overall 9780 ShakeMaps with magnitude 5.5 or greater. The 73 missing events therefore imply an error rate of 0.7 % which is in an acceptable range. Nevertheless, we can assume that the missing events would have on average lower shaking and impacts than the included events, since such events are more likely to get attention and therefore have a ShakeMap produced.

Another question is whether the ShakeMap coverage is comparable across years. To answer this question we first need to consider the reference data. As we already discussed before, the ComCat list is most likely not as complete for the time period 1960-1972, as for the years after that. In Figure C3 we can see that the total number of events for "all earthquakes" (ShakeMap events plus ComCat events without ShakeMap) seemingly increases over time. However, this should in theory not be the case. The number of earthquakes per year should be more or less constant over years. Part of the variation is natural noise, but a lot

of the increase can probably be explained by missing events for the time period 1960-1972. This becomes particularly apparent when events below magnitude 5.5 are considered. Another possible reason for the increase is that duplicates in the ComCat database are more likely for the time periods with more data contributing networks. Since additional networks were added over the years, the number of duplicate events might also increase with years. In particular the year 2011 looks suspiciously like it might hold a large number of duplicate events. Not only because the number of events is exceptionally large, but also because

the number of events with magnitude 5.5 and without ShakeMap is surprisingly high for such a recent year. This needs to be kept in mind when comparing the number of ShakeMap events with the number of all events in Figure C3. Nevertheless, we can clearly see that from 1960 to 1970 ShakeMaps only exist for very few selected events which all have fatalities. Starting in 1970 ShakeMaps generally exist and are systematically produced from 1973 onwards. In 2006/2007 the share of events with ShakeMaps drastically increases.

The ShakeMap data is increasingly complete in more recent years and it is particularly incomplete before 1973 (and even more so before 1970). The extra ShakeMaps in more recent years, however, come from more and more weaker events receiving a ShakeMap. We can also confirm this with the distribution of magnitude for events with and without ShakeMap in Figure C4. With increasing magnitude also the share of ShakeMaps increases. Most high magnitude events do have a ShakeMap and the lower magnitude (but above 5.5) events without ShakeMap are on average older events than those with ShakeMap. This Figure

also confirms that "big" events do generally have a ShakeMap and we are not missing a significant number of high magnitude events in the ShakeMap dataset.

A bigger concern is whether the ShakeMaps have a geographic bias. Since the USGS is a North American institution, we expect that ShakeMaps for low magnitude earthquakes in North America are more likely produced than for such events from different regions of the world. Comparing global maps of the number of events in the ComCat list to the number of ShakeMaps does indeed confirm a bias towards more North American events, in particular along the US West Coast. Figure C5 helps us to investigate the North America bias further. We can loosely define an event to be in North America, if its epicenter has a latitude between -170 and -60, and a longitude between 25 and 70. We can then see that for all events since 1973, almost all ShakeMaps with magnitude under 4 are from North America and there is a strong bias towards North American events until about magnitude 4.5. Between magnitude 4.5 and 5.5 still relatively more North American ShakeMaps are available, but the bias is in a reasonable range. For events with magnitude greater than 5.5 no apparent North American bias exists. The left panel in Figure C5 has some outliers with a much lower than expected ratio of high magnitude events having a ShakeMap. The most pronounced outliers are even for the share of North American events being low for some relatively high magnitude values. It is unlikely that there are actually that many high magnitude events (particularly in North America) without ShakeMap and we are most likely seeing the effect of duplicates in the ComCat list artificially increasing the denominator.

It is advised to restrict the dataset to ShakeMaps with magnitude above 4.5 or even 5.5 for many applications to avoid the North America bias. Events with a magnitude under 5.5 can still occasionally cause severe impacts. We therefore don't want to exclude all of them, particularly since the events often have a ShakeMap if they did indeed cause significant impacts. Nevertheless, it is important to be aware that the sample is geographically biased for earthquakes below the magnitude threshold of 5.5. If we only consider ShakeMaps from events with a magnitude of 4.5 or higher our sample size will be 12388.

## Appendix D:  Calculation of the shaking center

In the sample of 11510 ShakeMaps with positive shaking and magnitude 4.5 or higher, 10401 ShakeMaps have a unique maximum PGA location, 656 events have two grid cells with the maximum PGA, and 453 ShakeMaps have more than two grid cells sharing the maximum PGA. For the earthquakes with more than one grid cell as potential shaking center, it is necessary to define a consistent way to pick one of them as the shaking center. The here applied approach to tackle this problem is to incrementally add the surrounding cells of the cell and calculate the average shaking value in that square. Only those shaking center candidates are then kept that reach the highest value for that measure, until only one location remains. This way location with the maximum shaking, that has the strongest shaking in the area surrounding it, is chosen as the shaking center. This procedure reaches in only 47 cases with PGA as the ground motion parameter the edge of the ShakeMap. We then assume that the average shaking outside of the ShakeMap is the same as the average shaking of the added cells that are still in the ShakeMap at the same distance to the potential shaking center. However, for 24 of the 47 events still no unique shaking center can be found, since they occurred in small island regions and only caused shaking in very small areas (those events have on average only 17 km$^2$ exposed to any shaking). For those events the location closest to the shaking centroid which experiences

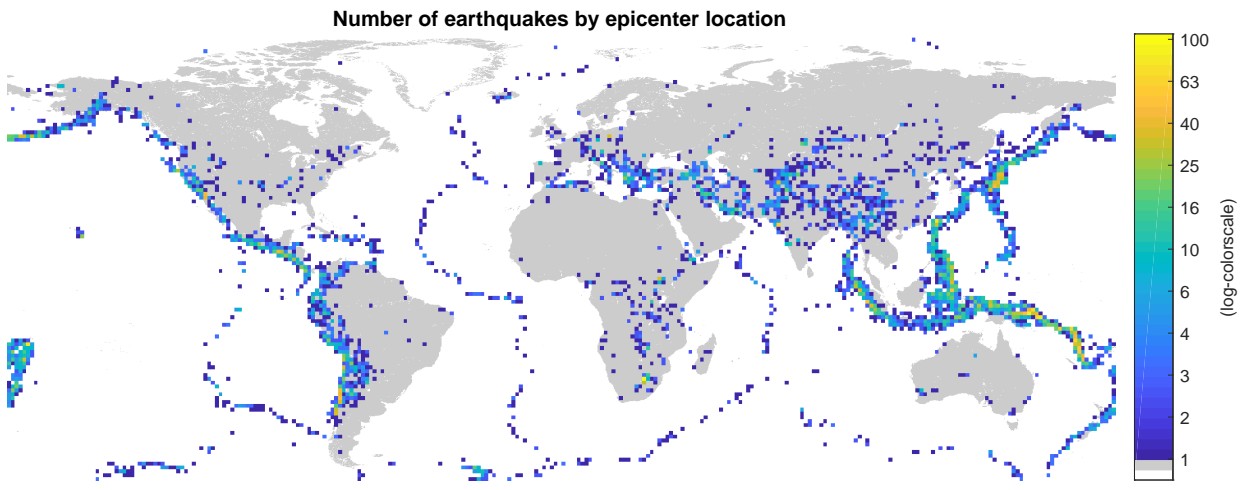

**Figure C1.** Epicenter locations of the sample of 12 388 earthquakes in the dataset with ShakeMaps and magnitude 4.5 or greater. The map shows the number of earthquakes in the dataset which have their epicenter in the corresponding 1.25 x 1.25 degree grid cell.

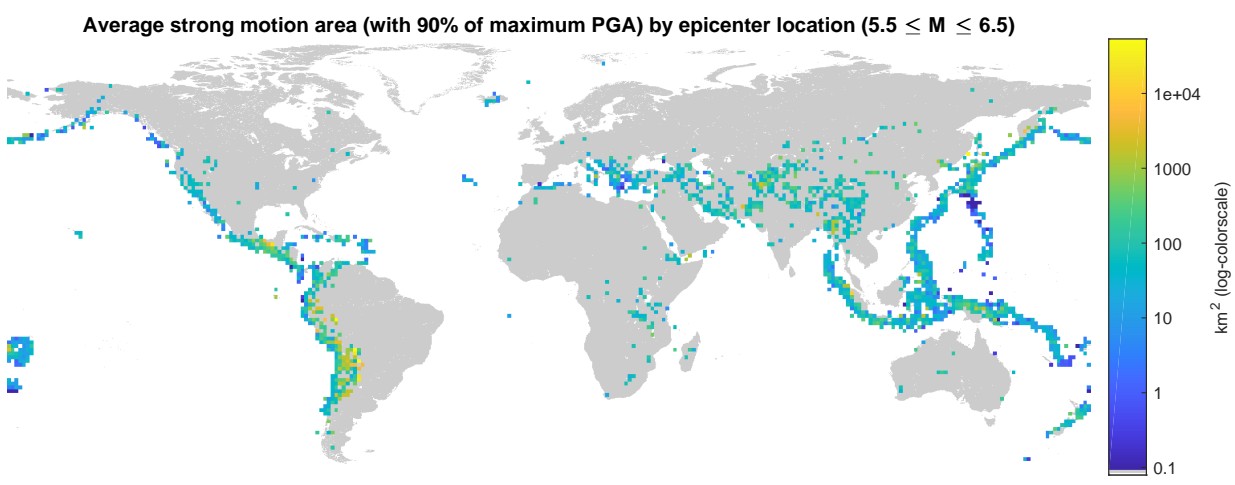

**Figure C2.** Attenuation across the world illustrated by average strong ground motion area for 1973-2015 and earthquakes with magnitude between 5.5 and 6.5. This map shows the average area that is exposed to at least 90 % of the maximum PGA for an earthquake with its epicenter at that location.

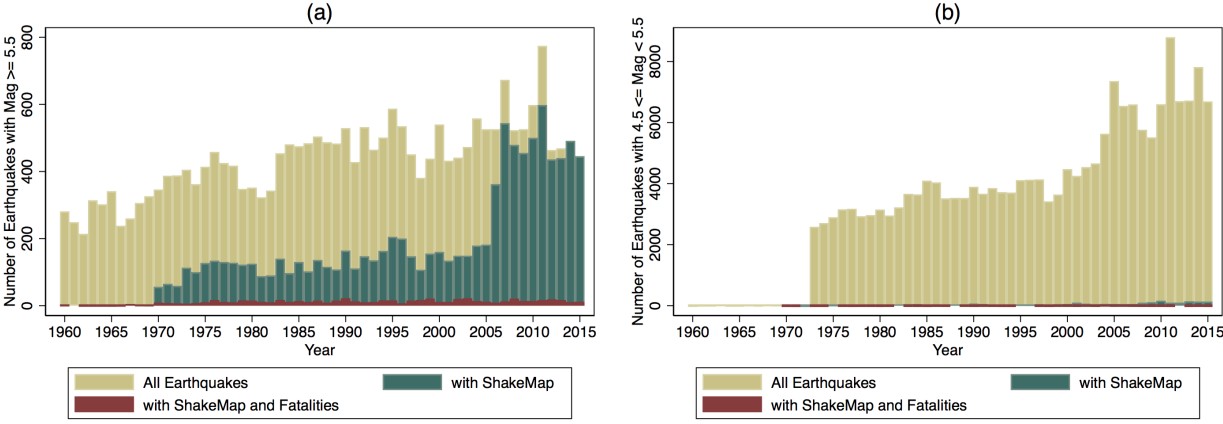

**Figure C3.** Number of earthquakes per year for given magnitude ranges. Before 1973 ComCat has a significant number of missing events, particularly for events with magnitude below 5.5. Between 1960 and 1970 ShakeMaps only exist for individual events with (high) fatalities. Starting 1973 ShakeMaps are produced systematically, and from 2007 onwards almost all events with magnitude 5.5 or greater have a ShakeMap.

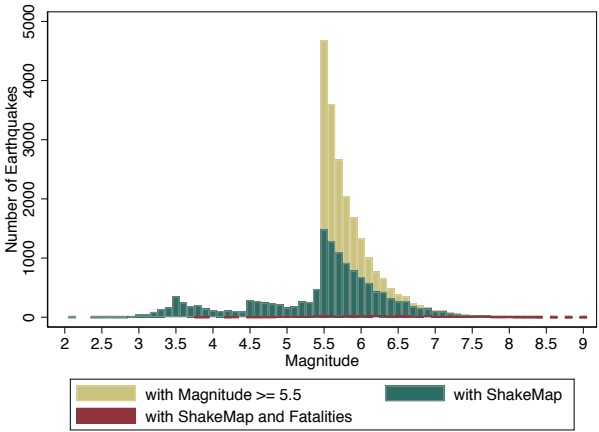

**Figure C4.** The distribution of magnitude for events from January 1973 to October 2016. ShakeMaps are systematically produced for all events with magnitude 5.5 or greater and the share of events with ShakeMap increases with magnitude.

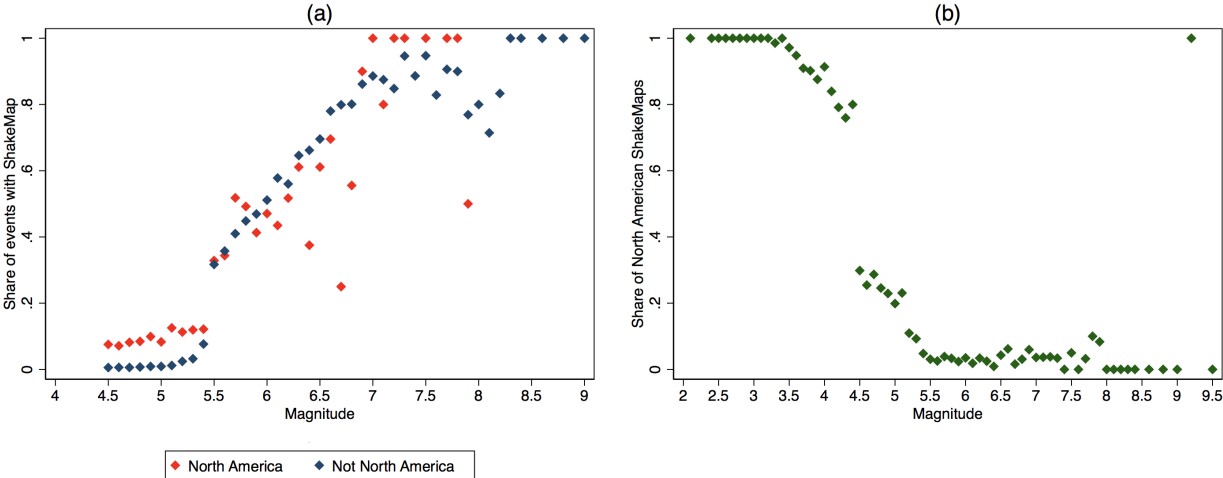

**Figure C5.** The share of events with ShakeMap - panel (a) - and the share of North American ShakeMaps - panel (b) - by magnitude for events since 1973. Below magnitude 5.5 ShakeMaps for North American events are more commonly produced, and below magnitude 4.5 they are almost exclusively produced for North American events.

maximum shaking is chosen as the shaking center. This procedure results in a unique shaking center for all 11510 events. In case that this would not be sufficient, the smallest distance to the epicenter can be considered.

## Appendix E:  Comparison of earthquake shaking history with GSHAP hazard data

ShakeMap data can be used to compare probabilistic shaking estimates for the future (hazard maps) with actual shaking
5   occurrences. To illustrate this the here calculated earthquake maximum shaking history (maximum PGA exposure for 1973-2015) is compared with the GSHAP global earthquake hazard map in Figure E1. The scale is cut off at a difference of 10 %g, presenting any higher values in the same color as a difference of exactly 10 %g.

Since only limited documentation about the GSHAP data could be found, it is necessary to make some assumptions about the data before it can be combined with the ShakeMap data. In particular: (i) it is assumed that the coordinates refer to the
10   center of each gridcell, and (2) the data for the longitude 193 is dropped due to repetition of the longitude column (longitude -167 is kept in the data). For this comparison, the maximum shaking history is calculated at the resolution of the GSHAP data of 1/10 x 1/10 degrees. Each gridcell is assigned the maximum PGA value exposure that occurred anywhere in that gridcell at some point between 1973-2015 according to the ShakeMap data.

*Competing interests.*  The author declares that no competing interests are present.

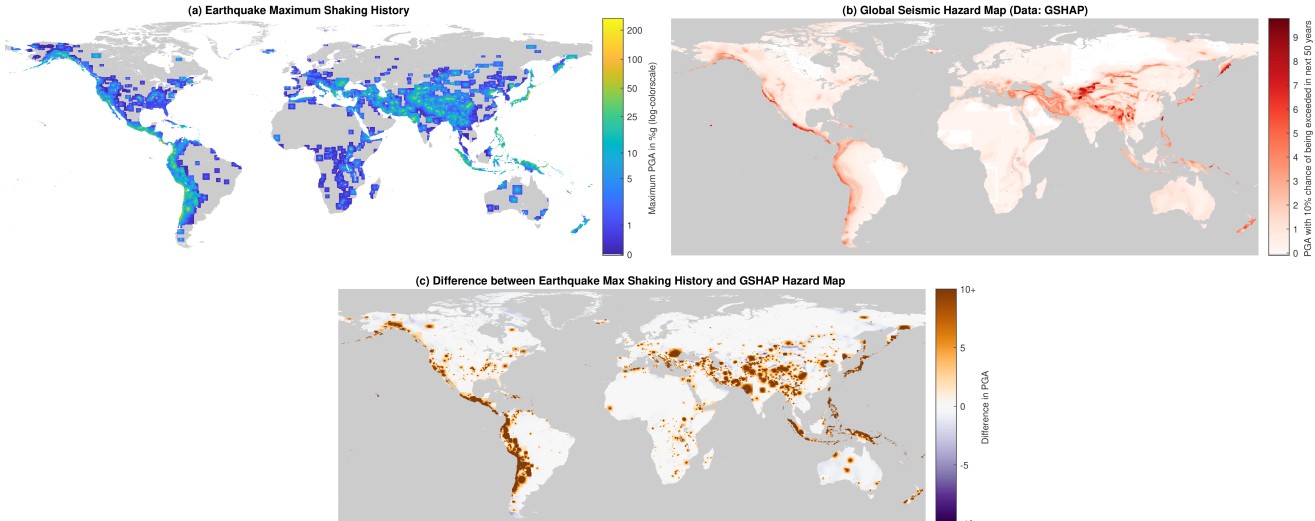

**Figure E1.** Comparing the earthquake maximum shaking history with the GSHAP hazard map. Panel (a) shows the maximum shaking experienced between 1973-2015, and panel (b) plots the GSHAP data of probabilistic PGA estimates that will not be exceeded with a 90 % chance within 50 years. Subtracting (b) from (a) results in the difference between actual exposure and the probabilistic estimates, which is displayed in panel (c). The scale is cut off at a difference of 10 %g. This comparison reveals that the maximum shaking history for the given time range tends to exceed the GSHAP estimates in most seismically active regions of the world. It is important to keep in mind that the shaking history is a combination of observed and estimated data for actual shaking exposure in the given time period.

*Acknowledgements.* I want to thank John Mutter, Art Lerner-Lam, Douglas Almond, Amir Jina, and George Deodatis for their advice on this work. I also thank Timothy Foreman, Jesse Anttila-Hughes, and Markus Riegler for helpful comments and discussions. I am grateful to David Wald and Michael Hearne for advice on the ShakeMap data and for providing some Matlab code that helped processing the data.

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

**Table 1.** Distances in km between the epicenter (EC), shaking center (SC), and shaking centroid (SCt), by whether the epicenter is in water or on land for all 11510 events with shaking and magnitude greater or equal 4.5.

| Epicenter | EC to SC | | | EC to SCt | | | SC to SCt | | |
|---|---|---|---|---|---|---|---|---|---|
| | Mean | StD | Max | Mean | StD | Max | Mean | StD | Max |
| On land (46 %) | 7 | 13 | 362 | 5 | 10 | 313 | 7 | 13 | 355 |
| In water (54 %) | 53 | 53 | 1022 | 59 | 54 | 682 | 25 | 33 | 445 |
| Total (100 %) | 32 | 46 | 1022 | 35 | 48 | 682 | 16 | 28 | 445 |

**Table 2.** Average area in km$^2$ exposed to at least 90 % of the maximum PGA, by magnitude and maximum PGA level.

| Magnitue | PGA (%g) | | | | | |
| --- | --- | --- | --- | --- | --- | --- |
| | 0-10 | 10-20 | 20-40 | 40-80 | >80 | Total |
| <4.5 | 16 | 11 | 14 | 14 | | 15 |
| 4.5-5.5 | 149 | 63 | 30 | 31 | 22 | 100 |
| 5.5-6 | 332 | 78 | 63 | 34 | 41 | 180 |
| 6-6.5 | 544 | 106 | 89 | 74 | 44 | 234 |
| 6.5-7 | 714 | 237 | 237 | 150 | 81 | 316 |
| 7-7.5 | 542 | 31 | 609 | 352 | 117 | 439 |
| 7.5-8 | 2448 | 67 | 589 | 456 | 230 | 675 |
| ≥8 | 2798 | 99 | 423 | 396 | 128 | 556 |
| Total | 255 | 87 | 133 | 151 | 101 | 171 |

**Table 3.** Average area in 100 km$^2$ exposed to at least 10 %g PGA, by magnitude and maximum PGA level.

| | PGA (%g) | | | | |
|---|---|---|---|---|---|
| Magnitude | 10-20 | 20-40 | 40-80 | >80 | Total |
| <4.5 | <1 | 4 | 6 | | 3 |
| 4.5-5.5 | 3 | 6 | 17 | 19 | 4 |
| 5.5-6 | 6 | 14 | 22 | 36 | 9 |
| 6-6.5 | 8 | 25 | 43 | 62 | 17 |
| 6.5-7 | 30 | 63 | 75 | 135 | 56 |
| 7-7.5 | 8 | 145 | 154 | 247 | 136 |
| 7.5-8 | 12 | 199 | 251 | 552 | 242 |
| ≥8 | 9 | 186 | 710 | 685 | 443 |
| Total | 7 | 35 | 87 | 221 | 25 |

**Table A1.** Number of ShakeMaps in the dataset by magnitude and maximum PGA level. The magnitude distribution of earthquakes with magnitude above 5.5 follows a power law, as expected for a comprehensive sample of earthquakes. For lower magnitude events ShakeMaps are not consistently produced and a less consistent magnitude distribution is therefore observed in that range.

| Magnitude | PGA (%g) | | | | | | |
|---|---|---|---|---|---|---|---|
| | 0 | 0-10 | 10-20 | 20-40 | 40-80 | >80 | Total |
| <4.5 | 7 | 1593 | 242 | 277 | 101 | 0 | 2220 |
| 4.5-5.5 | 67 | 1263 | 853 | 390 | 34 | 1 | 2608 |
| 5.5-6 | 606 | 2070 | 2132 | 773 | 46 | 12 | 5639 |
| 6-6.5 | 156 | 716 | 888 | 649 | 72 | 21 | 2502 |
| 6.5-7 | 34 | 203 | 295 | 392 | 125 | 24 | 1073 |
| 7-7.5 | 10 | 48 | 51 | 189 | 68 | 33 | 399 |
| 7.5-8 | 4 | 15 | 12 | 47 | 41 | 14 | 133 |
| $\geq 8$ | 1 | 3 | 3 | 11 | 11 | 5 | 34 |
| Total | 885 | 5911 | 4476 | 2728 | 498 | 110 | 14608 |