# Peer review of "Earthquakes on the surface: earthquake location and area based on more than 14500 ShakeMaps"

_Natural Hazards and Earth System Sciences, 2017_

## Referee Comment (RC1) · Anonymous Referee #1 · 23 Feb 2018

The submitted paper presents a new approach to the topic of characterization of earthquake shaking trying to find new ways to its use in social sciences, insurances, etc. Earthquake hazard, cited many times on the text, mainly deals with the shaking of earthquakes "to come". Instead, this paper deals with the "past" shaking, giving some cues on how to analize and use such data for further research. It relies in the massive analysis of the information contained in the ShakeMaps acquired for more than 40 years by the USGS. It describes the used analysis methodology, introduces some new definitions with the aim of improving the analysis and presents some conclusions about the distribution of earthquake shaking around the world.

New, fresh approaches to problems are always welcome and as it the submitted research is really welcome. But, from another side, I think more in-depth knowledge of the topic of seismic hazard and risk, how it is assessed, communicated and managed is needed to draw proper conclusions. The common seismological terminology used and other existing approaches are not well presented and used.

Seismic hazard is a topic where, from the field of seismology, large efforts have been devoted. Specifically, and dealing with global approaches, on the 90's, during the UN/IDNDR, the GSHAP Global Seismic Hazard Map project was implemented. A large amount of information about it can be found in Giardini (1999) and in the (frozen) project website http://www.seismo.ethz.ch/static/GSHAP/index.html. One of the main outputs is a map depicting the global earthquake hazard as the 90% chance of non-exceedance in 50 years of plotted PGA values. As 50 years is the approximate period covered at present by the ShakeMaps it can be really instructive comparing the "true data" presented in this research with the much more classical probabilistic calculations obtained in the GSHAP. Nevertheless, it should be kept in mind that results plotted on the ShakeMaps are a mix of observed data and assumptions/calculations about how energy/shaking propagates. Thus, the overall results are not just observed quantitative data. Anyway, I think such an exercise can show some of the possibilities the submitted research opens and it is adding value to the presented results.

At present many efforts in this topic (always from the fields of seismology and earthquake engineering) are directed through the GEM project. How communicate seismic hazard and risk is an important issue on the project development. Much more information is available at the website https://www.globalquakemodel.org/.

Coming back to the submitted research, I think the final results about the shaking suffered around the globe, presented in a 2.5° x 2.5° world partitioning, are not so useful as such size is larger than usual administrative departments, districts, etc., and even of many not so big states. Even in case of many damaging earthquakes, such area exceeds the whole affected area (at least that where damage occurs) and just

in the case of the largest events can it be considered. Certainly, I think this is not a big deal as, if I understand properly, reducing the size of the area (e. g. 1.5° x 1.5°) just takes more computer memory and calculation time. As a résumé, I think the submitted paper presents new results, it is properly written and organized, with relevant bibliography. Thus I think the article should be accepted for publication but it needs revisions of some importance.

In addition to these general comments, I'm pointing in the next lines specific details.
* * *
Specific items.

Page 1. Line 5. . . . Earthquake communication outside of seismology. . . An (old) review about how this problem is seen by seismologist can be found at O'Brien and Mileti (2003).

Page 1. Line 7. . . .constructed and applied TO THE DETERMINATION OF GROUND SHAKING WORLDWIDE.

Page 1. Line 19. Practice. . . practical

Page 2. Line 17. (e. g. Shedlock et al., 2000)

Page 3. Line 2. Strong ground motion CAN BE EXPRESSED with different parameters, but. . .

Page 3. Line 5. But such data are not as. . .

Page 3. Line 16. I know it is really easy to find the USGS ShakeMaps, but it will be good to give an explicit link to the webpage.

Page 3. Line 27. As a result, the dataset.

Page 4. Line8. Earthquake shaking history is, to me, a good term; but it is not the first time such an approach is used. See the Italian macroseismic database where it

is possible to search the felt earthquakes at a town (https://emidius.mi.ingv.it/CPTI15-DBMI15/query_place/)

Page 4. Line 15. Second? There is not a "first". I understand the discussion in the previous paragraph is the "first" point. It is just a matter of some rewriting.

Page 5. Figure 3. Abscissa axis. The last number on the right does not fit.

Page 5. Line 4. . . .points to characterize earthquake LOCATION.

Page 5. Line 7. . . .is AT SEA.

Page 6. Line 6. (when it falls offshore)

Page 6. Line 13. When the epicenter lies offshore.

Page 7. Line 3. "This distance increases to 53 km when the epicenter is offshore". But this value is, for me, meaningless. It gives just a mean value of how distant from the coast are epicenters of earthquakes for which a ShakeMap is available. This value arises just because at the epicenter location there is no ground to shake. In fact, ShakeMap gives "calculated" PGA's in sea areas. Thus, it is possible to extent shaking centroid calculation at sea.

Page 8. Figure 6. I think this figure will improve if the color scale of scatter is added somewhere.

Page 8. Line 8. . . .when the epicenter lies offshore.

Pages 9 – 11. I suggest, for future research, to use the ISC catalogue instead of the ComCat. It has the advantage that it is accepted as "authoritave" catalogue for the world seismicity and that event duplications are almost totally filtered. Instead, it has the problem it is published with 2-3 years of delay from present.

Page 10. Lines 16 and 25. I think a rounding accuracy of one degree in latitude is too wide. Much likely reducing the interval to .5 (or .25) may solve some of the possible

duplicities.

Page 11. Line 10. . . .each event FROM the significant EARTHQUAKE list

Page 11. Line 17. . . .some events in the significant earthquake list seem to have typos.

Page 16. Line 8. For those events a "RANDOM" candidate. . . I suggest, instead of a "random" candidate, to select the shaking center nearest to the epicenter. Such a criterion has the advantage of "repeatability".

Bibliography

-Giardini, D. (Ed.) (1999). The Global Seismic Hazard Assessment Program (GSHAP): 1992-1999. Annali Geofis. 42(3-4), 272 pp.

-O'Brien, P. W. and D. S. Mileti (2003). The sociological Dimensions of Earthquake Mitigation, Preparedness, Response and Recovery. In Lee et al. (eds.): International Handbook of Earthquake and Engineering Seismology, Academic Press, Amsterdam, Part B, 1241-1252.

---

## Referee Comment (RC2) · Anonymous Referee #2 · 25 Feb 2018

This paper aims to provide a discussion of earthquake shaking for an interdisciplinary audience and with applications in earthquake impact research, particularly in the social sciences in mind. The author is not a seismologist but has a strong background in mathematics, natural disasters, their socio-economic impacts, and sustainable development. In other words, the main purpose of this paper seems to be an earthquake communication about global seismic hazard based on past events (of limited duration) in the social science literature.

A dataset of relevant global earthquake ground shaking from 1960 to 2016 based on USGS ShakeMap data has been constructed and applied in this paper, where it is

claimed to be the first global quantitative analysis on the size of the area that is on average exposed to strong ground motion (PGA), while introducing two new definitions of earthquake location (the shaking center and the shaking centroid) based on ground motion parameters.

The idea is great. However, the paper appears to be lacking in-depth knowledge of complexities in the earthquake rupture, seismic hazard and risk. References to local/site/geologic conditions and/or quality/seismic performance of buildings/structures have been kept to a minimum if none, and there are no discussions on the rupture heterogeneity and directivity and their relations to seismic hazard. Statements on earthquake rupture physics are generally very simplistic, such as "waves radiate out from every point of the rupture area", where there is no single reference to asperities and frequency content of the waves radiated and how these are connected to seismic hazard studies. The discussion on any relevant uncertainties seems to be kept to a minimum. The argument of "the literature commonly uses magnitude or other suboptimal measures to quantify the natural hazard of earthquakes for impact research" could be true for social sciences, but definitely not acceptable for a seismologist, and this paper does not appear to be improving this deficiency. It is not the "parameter" that is misleading in earthquake hazard communication, it is "how we communicate or not communicate" the parameters to the outside community. In that respect, the paper, despite its good intentions, does not necessarily serve the purpose set by the author.

The dataset is clearly biased, as also acknowledged by the author. While a magnitude threshold has been introduced by the author to remove the effect of geographical bias as much as possible, at least in Northern America, several geographical regions hosting seismotectonic settings resulting in catastrophic earthquakes with long recurrence intervals, such as Dead Sea Fault or Hellenic Arc, for example, are simply shown with low percentages of average annual maximum PGA (%g) exceeded as a result of no-big-earthquake record for the time interval considered in this paper. In addition to the issues related to the temporal analysis, the resolution of 2.5 x 2.5 degree grid cell is

most likely introducing spatial problems due to its large size, which may explain the geographical bias introduced in various regions as a result of large earthquakes during the study interval. The discussion on such uncertainties in the data set need to be clearly improved, since the figures of this paper, if communicated to non-seismology experts, may lead to convey misleading messages on the global seismic hazard and risk.

The author states that other factors (e.g. geology and water bodies) introduce significant noise in the relationship of magnitude and the area exposed to a particular shaking threshold as a result of this magnitude. This "noise", in reality, is the "data" for seismic hazard and risk studies, and impact assessment. The case of 7.4 1999 Izmit earthquake, for example, is a good example where extensive damage has occurred to high residential buildings in AvcÄślar-Istanbul, 120 km away from the epicenter, which is attributed to both the long distance effects of the high period waves of the earthquake and soil amplification.

It is questionable whether introducing two new definitions of earthquake location, namely the shaking center and the shaking centroid, respectively, will provide any additional benefit, both in terms of seismic hazard studies and earthquake communication. Earthquakes, even on the same fault with same/similar size, do not behave identically, and the well-established term "intensity" in seismology, as a measure of the strength of shaking produced by the earthquake at a certain location, is still not "digested" by many communities around the globe. In that respect, for example USGS' Pager is a considerable approach to link Modified Mercalli Scale with strong ground motion parameters, such as PGA and PGV, population exposed and potential damage.

In conclusion, the paper is an attempt with good intentions, where a publication is welcomed after enriching the paper by addressing (at least some of the) improvements implied in the comments given above. It may be very well that the author might have intentionally kept the technical/scientific details to a minimum, especially with relevance to the uncertainties in the physics of the phenomena and in the methodology, since the

target audience is particularly in the social sciences. However, the communication of such uncertainties specifically to these audiences is crucial in terms of global earthquake preparedness.

---

## Author Comment (AC1) · 24 Apr 2018

**Reply to RC1**

I would like to thank the reviewer for the thorough review and the many helpful comments and suggestions made to improve the present work. Please find below the reviewer's comments and author's replies to these comments.

**General comments**

**RC1:** *The submitted paper presents a new approach to the topic of characterization of earthquake shaking trying to find new ways to its use in social sciences, insurances, etc. Earthquake hazard, cited many times on the text, mainly deals with the shaking of earthquakes "to come". Instead, this paper deals with the "past" shaking, giving some cues on how to analize and use such data for further research. It relies in the massive analysis of the information contained in the ShakeMaps acquired for more than 40 years by the USGS. It describes the used analysis methodology, introduces some new definitions with the aim of improving the analysis and presents some conclusions about the distribution of earthquake shaking around the world.*

*New, fresh approaches to problems are always welcome and as it the submitted research is really welcome. But, from another side, I think more in-depth knowledge of the topic of seismic hazard and risk, how it is assessed, communicated and managed is needed to draw proper conclusions. The common seismological terminology used and other existing approaches are not well presented and used.*

**Reply:** I have now included many of the suggestions made by the reviewer below. In particular more details and references on the technical aspects of earthquakes have been added and the resulting limitations of the ShakeMap data are discussed.

**RC1:** *Seismic hazard is a topic where, from the field of seismology, large efforts have been devoted. Specifically, and dealing with global approaches, on the 90's, during the UN/IDNDR, the GSHAP Global Seismic Hazard Map project was implemented. A large amount of information about it can be found in Giardini (1999) and in the (frozen) project website http://www.seismo.ethz.ch/static/ GSHAP/index.html. One of the main outputs is a map depicting the global earthquake hazard as the 90% chance of non-exceedance in 50 years of plotted PGA values. As 50 years is the approximate period*

*covered at present by the ShakeMaps it can be really instructive comparing the "true data" presented in this research with the much more classical probabilistic calculations obtained in the GSHAP. Nevertheless, it should be kept in mind that results plotted on the ShakeMaps are a mix of observed data and assumptions/calculations about how energy/shaking propagates. Thus, the overall results are not just observed quantitative data. Anyway, I think such an exercise can show some of the possibilities the submitted research opens and it is adding value to the presented results.*

**Reply:** A short discussion of the GSHAP project has now been added as well as a comparison of the GSHAP data with the here constructed data set of shaking exposure (please see appendix of the paper).

**RC1:** *At present many efforts in this topic (always from the fields of seismology and earthquake engineering) are directed through the GEM project. How communicate seismic hazard and risk is an important issue on the project development. Much more information is available at the website https://www.globalquakemodel .org/.*

**Reply:** The GEM project is obviously an important international effort on many aspects of seismic hazard and risk communication. I have now added a reference to the project.

**RC1:** *Coming back to the submitted research, I think the final results about the shaking suffered around the globe, presented in a 2.5° x 2.5° world partitioning, are not so useful as such size is larger than usual administrative departments, districts, etc., and even of many not so big states. Even in case of many damaging earthquakes, such area exceeds the whole affected area (at least that where damage occurs) and just in the case of the largest events can it be considered. Certainly, I think this is not a big deal as, if I understand properly, reducing the size of the area (e. g. 1.5° x 1.5°) just takes more computer memory and calculation time.*

**Reply:** The 2.5 x 2.5 resolution was chosen to ensure a higher number of events within each gridcell for the calculation of the averages. The choice of the resolution makes a decision on how to balance spatial granularity vs. sample size and there is no obvious optimal choice. It is certainly possible to reduce the resolution, and I have now included the same figures for a resolution of 1.25 x 1.25 instead of 2.5 x 2.5.

**RC1:** *As a résumé, I think the submitted paper presents new results, it is properly written and organized, with relevant bibliography. Thus I think the article should be accepted for publication but it needs revisions of some importance.*

**Reply:** I want to thank the reviewer for their résumé and have included suggested revisions.

**Specific items**

**RC1:** *Page 1. Line 5. . . . Earthquake communication outside of seismology. . . An (old) review about how this problem is seen by seismologist can be found at O'Brien and Mileti (2003).*

**Reply:** My intention was to point out that any discussions/ conversations/ communication about earthquakes outside of seismology are often centered around concepts (e.g. magnitude and epicenter) that are not of direct concern for those cases (unlike surface shaking /strong ground motion). I was not trying to talk about earthquake communication as a part of disaster management in particular. I have rephrased this sentence to make this clearer. The referenced paper provides a great discussion of the social dimensions of earthquake disaster management (mitigation, preparedness, response, and recovery). However, it provides little discussion on the aspect and importance of earthquake shaking for these social dimensions and is thus in my opinion not a relevant

citation here.

**RC1:** *Page 1. Line 7. . . .constructed and applied TO THE DETERMINATION OF GROUND SHAKING WORLDWIDE.*

**Reply:** Added.

**RC1:** *Page 1. Line 19. Practice. . . practical*

**Reply:** Corrected.

**RC1:** *Page 2. Line 17. (e. g. Shedlock et al., 2000)*

**Reply:** Corrected. (I assume the comment was about line 1 and not 17.)

**RC1:** *Page 3. Line 2. Strong ground motion CAN BE EXPRESSED with different parameters, but...*

**Reply:** Changed sentence accordingly.

**RC1:** *Page 3. Line 5. But such data are not as. . .*

**Reply:** Corrected.

**RC1:** *Page 3. Line 16. I know it is really easy to find the USGS ShakeMaps, but it will be good to give an explicit link to the webpage.*

**Reply:** The link to the website is in the "Code and data availability" section.

**RC1:** *Page 3. Line 27. As a result, the dataset.*

**Reply:** Changed sentence accordingly.

**RC1:** *Page 4. Line8. Earthquake shaking history is, to me, a good term; but it is not the first time such an approach is used. See the Italian macroseismic database where it is possible to search the felt earthquakes at a town (https://emidius.mi.ingv.it/CPTI15-DBMI15/query_place/)*

**Reply:** As far as I can tell, the Italian macroseismic database provides an earthquake history at a town level, but does not provide a visualization on a pixel level that overlaps ShakeMaps (or comparable products) over time.

**RC1:** *Page 4. Line 15. Second? There is not a "first". I understand the discussion in the previous paragraph is the "first" point. It is just a matter of some rewriting.*

**Reply:** The "first" was on line 3 of page 4. In the new version on line 11.

**RC1:** *Page 5. Figure 3. Abscissa axis. The last number on the right does not fit.*

**Reply:** Fixed both axes. Thank you for spotting this mistake!

**RC1:** *Page 5. Line 4. . . .points to characterize earthquake LOCATION.*

**Reply:** Changed sentence accordingly.

**RC1:** *Page 5. Line 7. ...is AT SEA.*

*Page 6. Line 6. (when it falls offshore)*

*Page 6. Line 13. When the epicenter lies offshore.*

*[...]*

*Page 8. Line 8. . . .when the epicenter lies offshore.*

**Reply:** I have intentionally avoided to use "offshore" or "at sea" since the epicenter could also fall into a lake. I am aware that this is a scenario of very low relevance, but I didn't want to exclude the theoretic possibility.

**RC1:** *Page 7. Line 3. "This distance increases to 53 km when the epicenter is offshore". But this value is, for me, meaningless. It gives just a mean value of how distant from the coast are epicenters of earthquakes for which a ShakeMap is available. This value arises just because at the epicenter location there is no ground to shake. In fact, ShakeMap gives "calculated" PGA's in sea areas. Thus, it is possible to extent shaking centroid calculation at sea.*

**Reply:** This is true, but the epicenter is used as a spatial reference for earthquakes no matter if the epicenter is offshore or not. When it is offshore it is a worse spatial reference for where shaking on land (which is what most people care about) is happening. This number helps to describe how much worse on average it is as a spatial reference.

ShakeMaps do provide calculated PGA's in sea areas, but they are not intended to be used since they are not based on models with the same level of quality as the estimates on land.

**RC1:** *Page 8. Figure 6. I think this figure will improve if the color scale of scatter is added somewhere.*

**Reply:** Color scales have now been added for the individual subplots of the figure. Since the color is just used to illustrate the density of the scatters using a shared color

scale would reduce the usefulness of using color in the first place.

**RC1:** *Pages 9 – 11. I suggest, for future research, to use the ISC catalogue instead of the ComCat. It has the advantage that it is accepted as "authoritave" catalogue for the world seismicity and that event duplications are almost totally filtered. Instead, it has the problem it is published with 2-3 years of delay from present.*

**Reply:** Thank you for the suggestion on improving future research.

**RC1:** *Page 10. Lines 16 and 25. I think a rounding accuracy of one degree in latitude is too wide. Much likely reducing the interval to .5 (or .25) may solve some of the possible duplicities.*

**Reply:** The rounding of the listed variables is only done to identify each event uniquely in the ComCat list, after 39 events were excluded from the dataset because they were identified as very likely duplicates through manual comparison. The stated rounding rules do explicitly result in uniquely identifiable events (no duplicates identifiable). However, I explain in Appendix C that I nevertheless believe that there are still duplicate events in the database (even though these rounding rules make it appear as if there shouldn't be any).

The matching of the events between the three data sets is only based on the rounded values in the first step when a unique match can be achieved. In the later matching steps the values as reported in ComCast are used. Changing the rounding accuracy, can therefore only change at what step events might be matched and not how or if they would be matched.

**RC1:** *Page 11. Line 10. . . .each event FROM the significant EARTHQUAKE list*

**Reply:** Corrected.

**RC1:** *Page 11. Line 17. . . .some events in the significant earthquake list seem to have typos.*

**Reply:** Changed sentence accordingly.

**RC1:** *Page 16. Line 8. For those events a "RANDOM" candidate. . . I suggest, instead of a "random" candidate, to select the shaking center nearest to the epicenter. Such a criterion has the advantage of "repeatability".*

**Reply:** The suggestion has been slightly adjusted and included. The algorithm selects now the shaking center nearest to the shaking centroid. This solves all the not-unique cases in the data. In case that this would not be sufficient, the smallest distance to the epicenter can be considered. However, this does not solve that theoretically it is still possible to not have a unique shaking center.

---

## Author Comment (AC2) · 24 Apr 2018

**Reply to RC2**

I would like to thank the reviewer for the comments and suggestions made to improve the present work. Please find below the reviewer's comments and author's replies to these comments.

**RC2:** *This paper aims to provide a discussion of earthquake shaking for an interdis-*

[Figure]

*ciplinary audience and with applications in earthquake impact research, particularly in the social sciences in mind. The author is not a seismologist but has a strong background in mathematics, natural disasters, their socio-economic impacts, and sustainable development. In other words, the main purpose of this paper seems to be an earthquake communication about global seismic hazard based on past events (of limited duration) in the social science literature.*

*A dataset of relevant global earthquake ground shaking from 1960 to 2016 based on USGS ShakeMap data has been constructed and applied in this paper, where it is claimed to be the first global quantitative analysis on the size of the area that is on average exposed to strong ground motion (PGA), while introducing two new definitions of earthquake location (the shaking center and the shaking centroid) based on ground motion parameters.*

*The idea is great. However, the paper appears to be lacking in-depth knowledge of complexities in the earthquake rupture, seismic hazard and risk. References to local/site/geologic conditions and/or quality/seismic performance of buildings/structures have been kept to a minimum if none, and there are no discussions on the rupture heterogeneity and directivity and their relations to seismic hazard. Statements on earthquake rupture physics are generally very simplistic, such as "waves radiate out from every point of the rupture area", where there is no single reference to asperities and frequency content of the waves radiated and how these are connected to seismic hazard studies. The discussion on any relevant uncertainties seems to be kept to a minimum. ...*

*[...]*

*It may be very well that the author might have intentionally kept the technical/scientific details to a minimum, especially with relevance to the uncertainties in the physics of the phenomena and in the methodology, since the target audience is particularly in the social sciences. However, the communication of such uncertainties specifically to*

*these audiences is crucial in terms of global earthquake preparedness.*

**Reply:** Yes, the discussion of the complexities of earthquake rupture processes and other scientific details on earthquakes that are relevant for seismic hazard and risk have intentionally been kept to a minimum. I do agree that acknowledging these factors is important. However, a detailed discussion of all the factors that play a role for what levels of strong ground motion manifests on the surface does not fit in the context of this paper. The primary concern is that social scientist need to be aware of the general limitations of the data due to the connected uncertainties. Earlier versions of this work actually included more technical details, which have been reduced in an attempt to make it easier readable for a social science audience. This reduction might have reached beyond the intended objective and I have now added again some more technical details throughout the paper.

**RC2:** *The argument of "the literature commonly uses magnitude or other suboptimal measures to quantify the natural hazard of earthquakes for impact research" could be true for social sciences, but definitely not acceptable for a seismologist, and this paper does not appear to be improving this deficiency. It is not the "parameter" that is misleading in earthquake hazard communication, it is "how we communicate or not communicate" the parameters to the outside community. In that respect, the paper, despite its good intentions, does not necessarily serve the purpose set by the author.*

**Reply:** I absolutely agree that "how we communicate or not communicate" scientific parameters is of great importance. Nevertheless, despite many efforts from the scientific community, misunderstandings of what magnitude refers to do prevail among the general public. It is very common for people (without any seismological training) to assume that magnitude is either a measure for surface shaking or at least a good proxy for it (some evidence for this is demonstrated by ()). While seismologists (and earthquake engineers) do very well understand that this is not necessarily the case it is still normal to compare magnitude with impacts (e.g a scatter plot of magnitude vs.

fatalities in ()).

While it is important how a parameter is communicated, it also matters how often a parameter is communicated. Magnitude is a term that is well known (if not necessarily understood) by the general public. It is used in some form in essentially any public discussion and communication on earthquakes, even when it is not actually of direct relevance for the particular topic being discussed. Earthquake shaking on the other hand is often not included in discussions in which it would actually be a more meaningful parameter to consider than magnitude. In the social sciences (or public discourse, newspapers, etc.) magnitude is often used as "the parameter" for earthquake size, when actually a quantification of earthquake shaking is desired (e.g. ()). The mere ease of availability (in terms of being used to the parameter as well as the data for the parameter) of magnitude compared to quantifications of shaking, is an important contributing reason for why this happens.

**RC2:** *The dataset is clearly biased, as also acknowledged by the author. While a magnitude threshold has been introduced by the author to remove the effect of geographical bias as much as possible, at least in Northern America, several geographical regions hosting seismotectonic settings resulting in catastrophic earthquakes with long recurrence intervals, such as Dead Sea Fault or Hellenic Arc, for example, are simply shown with low percentages of average annual maximum PGA (%g) exceeded as a result of no- big-earthquake record for the time interval considered in this paper. In addition to the issues related to the temporal analysis, the resolution of 2.5 x 2.5 degree grid cell is most likely introducing spatial problems due to its large size, which may explain the geographical bias introduced in various regions as a result of large earthquakes during the study interval. The discussion on such uncertainties in the data set need to be clearly improved, since the figures of this paper, if communicated to non-seismology experts, may lead to convey misleading messages on the global seismic hazard and risk.*

**Reply:** This paper is not trying to make a claim on presenting a full picture of the seismic hazard potential in general, but merely document the data for what did happen (or partially what has been estimated to have happened) in the specific time range of 1973-2015. Any dataset restricted to a couple of decades will obviously suffer from the "temporal bias" of only actually representing that time range and missing events that didn't occur because of longer recurrence intervals. It would obviously be desired to solve this by including more data, which is unfortunately not possible at this point in time. Nevertheless, this is not the same as the data itself being biased in terms of misrepresenting what happened in the specified time range.

Turning to the representation of the data in Figure 5 and C2. The issue of the coarse resolution of 2.5 x 2.5 has also been raised by the first reviewer and they have now been updated at a resolution of 1.25 x 1.25. However, the choice of the resolution makes a decision on how to balance spatial granularity vs. sample size. The finer resolution reduces the sample size in each gridcell (see Figure C1 for reference) and therefore gives more weight to each individual event when calculating the averages. Thus making the Figures more sensitive to the "temporal bias" of only having data on a couple of decades. I have added more information on the uncertainties and limitations of the data in the captions of these figures as well as in the text.

**RC2:** *The author states that other factors (e.g. geology and water bodies) introduce significant noise in the relationship of magnitude and the area exposed to a particular shaking threshold as a result of this magnitude. This "noise", in reality, is the "data" for seismic hazard and risk studies, and impact assessment. The case of 7.4 1999 Izmit earthquake, for example, is a good example where extensive damage has occurred to high residential buildings in AvcAÌĹślar-Istanbul, 120 km away from the epicenter, which is attributed to both the long distance effects of the high period waves of the earthquake and soil amplification.*

**Reply:** I think this might be an instance where the term "noise" is used in different ways

in different disciplines. I do not mean to say that these factors introduce "random noise", the "noise" is obviously meaningful data. The point is that these factors make the relationship between magnitude and the size of the shaking area less straightforward. I have reformulated this paragraph to make this clearer, but I have kept the term noise, since I believe that it makes this point easier to understand for social scientists. I have also added some more information on the characteristics about an earthquake that matter for the manifestation of shaking on the surface in section 3 ("Past Earthquake Shaking").

**RC2:** *It is questionable whether introducing two new definitions of earthquake location, namely the shaking center and the shaking centroid, respectively, will provide any additional benefit, both in terms of seismic hazard studies and earthquake communication. Earthquakes, even on the same fault with same/similar size, do not behave identically, and the well-established term "intensity" in seismology, as a measure of the strength of shaking produced by the earthquake at a certain location, is still not "digested" by many communities around the globe. In that respect, for example USGS' Pager is a considerable approach to link Modified Mercalli Scale with strong ground motion parameters, such as PGA and PGV, population exposed and potential damage.*

**Reply:** For certain applications in the social sciences as well as in earthquake communication being able to assign a singular location to an earthquake can be very crucial. In these cases the shaking locations would often be preferred compared to using the epicenter. A particular example that I provide in the conclusion is the use of the shaking centroid for spatial regression models or other statistical tools that require coordinates for each event. Using the epicenter for such applications can introduce significant noise into what the location is intended to represent.

Shaking intensity, while having its own problems, is an important concept that (just as instrumental quantifications of shaking) still often does not receive enough attention yet from some communities. The here introduced shaking locations can actually be complimentary to the concept of intensity in terms of bringing focus to earthquake shaking. The shaking location could even be calculated based on MMI data for locations with well established and consistent methods to generate MMI maps.

**References**

Celsi, R., Wolfinbarger, M., & Wald, D. (2005). The Effects of Earthquake Measurement Concepts and Magnitude Anchoring on Individuals; Perceptions of Earthquake Risk. Earthquake Spectra, 21(4), 987–1008. http://doi.org/10.1193/1.2099047

Bilham, R. (2010). Lessons from the Haiti earthquake. Nature, 463(7283), 878–879. http://doi.org/10.1038/463878a

Noy, I. (2009). The macroeconomic consequences of Disaster. Journal of Development Economics, 88(2), 221–231. http://doi.org/10.1016/j.jdeveco.2008.02.005

---

## Editor Decision (ED1)

Corrections suggested by the Editor on the revised manuscript:

Page 4 line 28 – insert a comma after "In a similar way" – "In a similar way,"
Page 4 line 33 – correct "in panel (a)" to in Figure 1 panel (a)
Page 5 line 2 - correct "in panel (b)" to in Figure 1 panel (b)
Page 10 line 4 – correct (see Figure C1 for reference) to see Figure C1 in appendix C for reference
Page 10 line 19 – Please clarify what is meant by – everything else fixed. Please insert what are the parameters that are fixed to make the reading of the text easier.
Page 11 line 1 – I believe that the first line should read "This study provides a discussion of earthquake shaking for …" should read "This study provides a discussion of earthquake shaking data for

---

## Author Response (AR2)

**Submission of revised manuscript**

Dear editor,

I am submitting my revised manuscript with the title "Earthquakes on the surface: earthquake location and area based on more than 14500 ShakeMaps".

Please find a list of all relevant changes below. Attached in this document are also your suggested changes and a marked-up manuscript version (created with latexdiff). Since I have changed the labels for all figures and tables in the LaTeX file, it highlights unfortunately every reference to a figure as a change, even though there was no change.

Kind regards,

Stephanie Lackner

**List of relevant changes**

- I have included all your suggest changes.
- In the "data and code availability" section I have added a link to a GitHub repository with all the code that was used for this research and additional information about the raw data.
- Another tick on the x-axis of Figure 3 was added.
- To make the work more transparent and reproducible, I have changed the code to be able to run from one runall.sh shellfile. This meant that I had to slightly change how some figures were produced, and they have now some minor differences to before:
  - The panel labelling has been standardized across figures to be centered above the panel (this affects Figures C3 and C5).
  - Figures 4, and C3-C5 were previously produced with Stata 13, but now with Stata 15. This caused slight visual changes (e.g. the tiny bars in the histograms are differently rendered).

 To comply with the NHESS requirement, I have now combined panels in one figure into one image. This has slightly changed the resolution for the figures with several panels.

**NHESS nhess-2017-422 Earthquakes on the surface: earthquake location and area based on more than 14500 ShakeMaps by Stephanie Lackner**

Corrections suggested by the Editor on the revised manuscript:

Page 4 line 28 - insert a comma after "In a similar way" - "In a similar way,"

Page 4 line 33 – correct "in panel (a)" to in Figure 1 panel (a) Page 5 line 2 - correct "in panel (b)" to in Figure 1 panel (b)

Page 10 line 4 – correct (see Figure C1 for reference) to see Figure C1 in appendix C for reference Page 10 line 19 – Please clarify what is meant by – everything else fixed. Please insert what are the parameters that are fixed to make the reading of the text easier.

[revised manuscript text omitted]